# Foliar Traits Drive Chlorophyll Fluorescence Variability in Chilean Sclerophyllous Species Under Early Outplanting Stress

**DOI:** 10.3390/plants14172682

**Published:** 2025-08-27

**Authors:** Sergio Espinoza, Carlos Magni, Marco Yáñez, Nicole Toro, Eduardo Martínez-Herrera

**Affiliations:** 1Departamento de Ciencias Forestales, Facultad de Ciencias Agrarias y Forestales, Universidad Católica del Maule, Talca 3460000, Chile; espinoza@ucm.cl; 2CESAF, Facultad de Ciencias Forestales y de la Conservación de la Naturaleza, Universidad de Chile, Santiago 8820808, Chile; crmagni@uchile.cl (C.M.); nitoro@uchile.cl (N.T.); 3College of Forestry, Agriculture, and Natural Resources, University of Arkansas at Monticello, 110 University Ct, Monticello, AR 71656, USA; yanez@uamont.edu

**Keywords:** evergreen and deciduous species, photochemical efficiency of photosystem II, chlorophyll a fluorescence, performance index

## Abstract

The photochemical efficiency of photosystem II (PSII) was monitored in two-year-old seedlings from six Chilean woody sclerophyllous species differing in foliage habits (evergreen, deciduous, semi-deciduous) and leaf orientation. A common garden experiment was established in July 2020 in a Mediterranean-type climate site under two watering regimes (2 L^−1^ seedling^−1^ week^−1^ for 5 months versus no irrigation). Chlorophyll a fluorescence rise kinetics (OJIP) and JIP test analysis were monitored from December 2021 to January 2022. The semi-deciduous *Colliguaja odorifera* (leaf angle of 65°) exhibited the highest performance in processes such as absorption and trapping photons, heat dissipation, electron transport, and level of photosynthetic performance (i.e., parameters PI_ABS_ F_V_/F_M_, F_V_/F_0_, and ΔV_IP_). In contrast, the evergreen *Peumus boldus* (leaf rolling) exhibited the opposite behavior for the same parameters. On the other hand, the deciduous *Vachelia caven* (small compound leaves and leaf angle of 15°) showed the lowest values for minimal and maximal fluorescence (F_0_ and F_M_) and the highest area above the OJIP transient (S_m_) during the study period. Irrigation decreased S_m_ and the relative contribution of electron transport (parameter ΔV_IP_) by 22% and 17%, respectively, but no clear effects of the irrigation treatments were observed among species and dates of measurement. Overall, *V*. *caven* and *C*. *odorifera* exhibited the highest photosynthetic performance, whereas *P*. *boldus* seemed to be more prone to photoinhibition. We conclude that different foliar adaptations among species influence light protection mechanisms more than irrigation treatments.

## 1. Introduction

Climate change represents a fundamental challenge for the sustainability of sclerophyllous vegetation in Mediterranean-type climate areas where multiple stresses, such as drought, high irradiation, and temperature, usually co-occur during summer and are expected to exacerbate in the coming decades [1]. New strategies will be required to ensure the recruitment and/or successful restoration of sclerophyllous vegetation in these areas. The identification of appropriate species, with adaptations that guarantee fast and successful establishment under severe growing conditions, is thus a priority task. One possible means to assess this task is the implementation of common garden trials to test the suitability of different species to be grown under harsh ecological conditions [2] and the use of a reliable and non-destructive method that can rapidly assess photosynthetic efficiency and the plant response to planting stress (e.g., the OJIP-test [3,4]).

The OJIP-test analyzes the fate of photons absorbed by PSII antennae (i.e., trapping, forward electron transport beyond QA, and dissipation as heat [5]). This test has been successfully used as a tool to detect plant stress conditions in Mediterranean species submitted to different stresses [6,7,8,9,10]. Most studies generally use the maximum quantum yield of primary PSII photochemistry (F_V_/F_M_) as the main parameter to assess stress conditions. However, the incorporation of parameters such as the performance index (PI), which synthesizes all the events of photon trapping and the photochemical phase, the probability that a photon trapped by the PSII reaction center enters the electron transport chain (ψE_0_), the efficiency of the water-splitting system (F_V_/F_0_), and the efficiency of electron flux through PSI (ΔV_IP_) [11], provides more information. This information can help monitor alterations in the photosynthetic apparatus to a wide range of stressors in terms of absorption and trapping photons, dissipation of excess energy, electron transport, and the level of photosynthetic performance [12,13].

Sclerophyllous species of Mediterranean-climate habitats have several leaf adaptations to survive in arid environments characterized by water limitations and high radiation loads. Steeply inclined leaves (45° or higher) are typically found in sclerophyllous species as they help to reduce exposure to excess light levels during periods of high radiation that can cause photoinhibition (i.e., sun at midday) and maintain photosynthesis despite intercepting less light [14,15,16,17,18]. However, the mechanisms of light protection after early planting in sclerophyllous species of the Mediterranean semi-arid environments of Central Chile, a highly degraded ecosystem that covers ca. 2 million hectares [19], remain unclear. This ecosystem harbors open scrub vegetation with spiny shrubs, sub-shrubs, and small trees that develop under restricted growth conditions. Species found in this ecosystem differ in foliar habits (i.e., evergreen, semi-deciduous, and deciduous species) and leaf orientations (i.e., steeply inclined angles, leaf folding, leaf rolling) as strategies to adapt to dry environments [20,21]. Still, the implications for light capture and photosynthetic performance have not been quantified. In addition to high radiation loads experienced by plants in these ecosystems, exposure to low water availability is also an important environmental stress affecting photosynthesis and chlorophyll fluorescence parameters, e.g., refs. [22,23]. However, results indicate increased activity of the photochemical apparatus [24] but also reduced PSII photochemical activity [25] under water scarcity. Similar to the influence of leaf adaptations on chlorophyll fluorescence parameters, the effect of low water availability on chlorophyll fluorescence parameters has not been evaluated in Chilean sclerophyllous species.

In the present contribution, we report on experimental plantations of sclerophyllous species from Central Chile (33° to 38° S), i.e., *Vachelia caven* (Molina) Seigler & Ebinger, *Lithraea caustica* (Mol.) Hook. and Arn, *Peumus boldus* Molina, *Colliguaja odorifera* Molina, *Escallonia pulverulenta* (Ruiz and Prav.) Pers, and *Quillaja saponaria* Molina, established in a common garden under typical growth conditions of the Mediterranean-type climate areas. These species were considered particularly appropriate for this study because they are the most common species occupying the sclerophyllous formations in Central Chile [26] and because of the high diversity of life forms, leaf habits, and leaf angles they exhibit. We aimed to analyze how different foliar adaptations among species and irrigation treatments influence light protection mechanisms during the summer months in a site with a Mediterranean-type climate.

## 2. Results

### 2.1. Temporal Variations in F_0_ and F_M_ Associated with the Species and Watering Treatments

F_0_ and F_M_ exhibited temporal variations associated with the species and watering treatments, but these variations were mainly attributed to the effect of the species and date of measurement (F-values in Table 1). Consistently, the species *V*. *caven* (shrub, deciduous) exhibited the lowest values of F_0_ and F_M_ in both the W+ and W− treatments during the study period, and these values were on average 16% lower in the W+ treatment (Figure 1). F_0_ was lower in *V*. *caven* in early December in the W+ treatment, whereas this parameter was higher in early January in *P*. *boldus* (tree, evergreen) in the W− treatment (Figure 1A,B). By the end of the study period, F_0_ remained low in *V*. *caven* and high in *P*. *boldus* in both the W+ and W− treatments. In the case of FM, *V*. *caven* exhibited the lowest values both in the W+ and W− treatments during the study period. From December 2021 to February 2022, the rest of the species exhibited a similar trend among them. There was an important decline in F_M_ during February 2022 (the hottest month of the study period) in most species. After that decline, most species recovered to values similar to those at the beginning of the experiment. However, *V*. *caven* did not recover in W−, and *P*. *boldus* had a pronounced slope in both W+ and W−. *Q*. *saponaria* exhibited high F_M_ during most of the experiment in both W+ and W−, but it experienced a decline in March 2022 (Figure 1C,D).

### 2.2. Responses of the Chlorophyll Fluorescence Parameters of the Different Species According to the Watering Treatment

Higher and lower values of F_V_/F_M_ and F_V_/F_0_ were found in *C*. *odorifera* (shrub, semi-deciduous) and *P*. *boldus,* respectively, but the interactive effects of species and watering were mainly associated with the differences exhibited by *L*. *caustica* (tree, evergreen) in both W+ and W− (Table 2, Figure 2A,B). In *L*. *caustica,* both F_V_/F_M_ and F_V_/F_0_ were 7 and 21% lower in the W+ treatment in comparison to the W− treatment. In the case of PI_ABS_, the differences among the species in both watering treatments were associated with the responses observed in *L*. *caustica* and *P*. *boldus,* which exhibited the opposite response. Higher PI_ABS_ for *L*. *caustica* was reported in W− (53% higher), whereas in *P*. *boldus,* PI_ABS_ was 50% higher in W+ (Figure 2C). For ψE_0_, differences among the species and watering were also found in *L*. *caustica* and *P*. *boldus*, but they were more accentuated in *P*. *boldus*. ψE_0_ in this species was 17% higher in W+ compared to W− (Figure 2D). Surprisingly, *L*. *caustica* exhibited the highest and significant F_V_/F_M_, F_V_/F_0_, PI_ABS_, and ψE_0_ in W−.

### 2.3. Temporal Variations in Chlorophyll Fluorescence Parameters Among Species

Except for ABS/RC, the rest of the parameters were different among the species at the beginning of the experiment (Figure 3). In all the species, PI_ABS_, F_V_/F_M_, F_V_/F_0_, ψE_0_, and ΔV_IP_ were minimum in February 2022, which reported the highest temperature in the field trial during the experiment. During this month, PI_ABS_, F_V_/F_M_, F_V_/F0, ψE_0_, and ΔV_IP_ in *P*. *boldus* were 89, 44, 65, 31, and 70% lower in comparison to the values reported in early December 2021. By February 2022, only ABS/RC exhibited an increase of 62% in *P*. *boldus* (i.e., ABS/RC > 5) when the other species had an ABS/RC in the 2 to 2.5 range (Figure 3E). By the end of the experiment (i.e., 11 April 2022), parameters such as PI_ABS_ and ΔV_IP_ tended to recover in most of the species, where *C*. *odorifera* had the highest PI_ABS_, ΔV_IP_, ψE_0_, F_V_/F_M_, and F_V_/F_0_. The exception was *P*. *boldus*, whose values of all parameters (except ABS/RC) were among the lowest before February 2022. Still, after that month, the recovery rate of F_V_/F_M_, F_V_/F_0_, and ψE_0_ stood out in relation to the other species (Figure 3B,C,F).

### 2.4. Temporal Variations in Sm Among Species and Watering Treatments

The normalized area above the OJIP transient (S_m_) was the only variable that exhibited temporal variation associated with the species and the watering treatments (Table 3, Figure 4). The highest S_m_ was observed in *V*. *caven*, both in the W+ and W− treatments during February and March 2022, but it sharply decreased by the end of the experiment in April 2022 (Figure 4A,B). *P*. *boldus* exhibited the lowest S_m_ among the species up to February 2022 in both watering treatments, but after that date, it increased to a higher percentage than the other species (63 and 61% increase in W− and W+, respectively). Except for *V*. *caven* and *P*. *boldus*, by the end of the experiment in the W− treatment, the rest of the species showed an increased S_m_ by an average of 42% in comparison to the value reported in February 2022. However, in the W+ treatment, *L*. *caustica* increased by 67% in comparison to the average 33% increase of the other species (Figure 4B).

## 3. Discussion

### 3.1. Differences Among Species for Temporal Variations in Chlorophyll Fluorescence Parameters

In our study, almost all the species are considered drought-tolerant [34], but they have different adaptations to cope with high radiation, high temperature, and low water availability. *C*. *odorifera* has high leaf drop in summer [27]. *L*. *caustica* has thicker and denser leaves, whereas *V*. *caven* has small compound leaves. *P*. *boldus* has the thickest leaves and bears a high amount of trichomes [35]. The leaf absorptance varies from 78% for *Q*. *saponaria* to 87% for *L*. *caustica* [28], and the leaf angle varies from 15° to 65° (Table 3). The leaf angle controls the photon flux density that strikes the leaf surface and influences fluorescence parameters [16,18]. Steeper leaf angles decrease the susceptibility to photoinhibition [36]. Thus, the foliar adaptations of the species in our study seem to have influenced the temporal variations in chlorophyll fluorescence parameters differently.

*C*. *odorifera* is the only semi-deciduous species among the species under study, whereas *P*. *boldus* has the thickest leaves with trichomes and leaf rolling as a mechanism to decrease the transpiration rate [35]. *C*. *odorifera* showed higher values for PI_ABS_, F_V_/F_M_, F_V_/F_0_, and ψE_0_ than the rest of the species up to the hottest month of the experiment (February 2022), indicating that light trapping and electron transport beyond QA function better in this species. This is a probable consequence of leaf drop during extreme drought and a leaf angle of 65°, which is typical of species from interior desert communities, combined with leaf absorptance close to 80%, which could have decreased light absorption and protected leaves from excessive irradiance during summer. Both the leaf angle and leaf absorptance affect the amount of solar radiation absorbed by a leaf. Steeply inclined leaves help reduce exposure to excess light levels during periods of high radiation. In contrast, low leaf absorptance reduces leaf temperature by reducing the absorbed heat load on the leaf [17,37]. The higher F_V_/F_0_ (proxy of heat dissipation, [4]) reported in *C*. *odorifera* during the study period (Figure 3D) may explain the high photosynthetic capacity in PI_ABS_ of this species. On the contrary, PI_ABS_, F_V_/F_M_, F_V_/F_0_, and ψE_0_ were severely reduced in *P*. *boldus* during the experiment, but particularly during February 2022, in which F_V_/F_M_ decreased to values as low as 0.38 (Figure 3C), clearly indicating photoinhibition [38] and suggesting the typical down-regulation of PSII photochemistry considered as a photoprotective mechanism [39,40]. In general, F_0_ was minimum in *P*. *boldus* during January 2022, suggesting a temperature that can damage leaves. However, the reduction in F_V_/F_M_ during February did not seem to be caused by an increase in F_0_ (destruction of the reaction centers of PSII, becoming non-functional as energy traps); it was possibly due to the decline in F_M_ during this month in almost all species (Figure 1C,D).

Although *P*. *boldus* exhibits leaf folding, its relatively larger PSII antenna size (parameter ABS/RC) during February 2022 suggests the presence of a non-QA reducing reaction center [3,41] and corroborates the low efficacy in the photochemistry of PSII in the species. Under high temperatures, *P*. *boldus* seems to be unable to transfer electrons from the antennae complex to the reaction centers. Inactivation of reaction centers can occur because of closed stomata under water shortage conditions. Yáñez et al. [34] and Espinoza (unpublished) reported stomatal conductances as low as 0.05 mol H_2_O m^−2^ s^−1^ in potted and field-grown seedlings of *P*. *boldus* under conditions of water limitation. This suggests photoinhibition caused by high solar radiation, which is also concurrent with the higher air temperatures registered during February (Figure 5). Under these conditions, *P*. *boldus* closes its stomata and restricts the absorption of CO_2_, with a consequent decrease in the demand for products of the light phase of photosynthesis. This strict stomatal control under water stress conditions, together with a great ABS/RC (great antennae but few active reaction centers), implies that *P*. *boldus* was unable to transfer electrons beyond PSII after being absorbed by the light-harvesting complex. Consistently, ∆V_IP_ (i.e., the efficiency of reducing final acceptors beyond PSI, [42]) decreased markedly in *P*. *boldus* during February 2022 (Figure 3B). This parameter was always the lowest in *P*. *boldus* due to the reduction of the acceptor side of PSI from December to a minimum value in February, in which there was significant damage to PSI. By this date, all the species reduced ΔV_IP_, but *P*. *boldus* almost lost its capacity to reduce photochemically QA and to transfer electrons to the PSI acceptor side.

The highest S_m_ was found in *V*. *caven* in both W+ and W− (Figure 4). This species has small-sized compound leaves with a low inclination angle with respect to the horizontal. Small-sized compound leaves provide plants with a thin boundary layer that enhances heat dissipation, whereas in wider leaves, such as those of *Q*. *saponaria* or *E*. *pulverulenta* (with intermediate S_m_ during the experiment), leaf temperatures increase. Moreover, species with shallower-angled leaves intercept more light when the sun is at high angles in the sky, i.e., midday [17]. Thus, it seems that both functional traits provided *V*. *caven* the ability to close all RCs (high S_m_) with high excitation energy, which improved the amount of electron carriers reduced from F_0_ to F_M_ [4]. Consistently, *P*. *boldus* had the lowest S_m_ during the experiment, corroborating its low ability to pass electrons through the electron transport chain.

### 3.2. Responses of the Chlorophyll Fluorescence Parameters of the Different Species According to the Watering Treatment

Environmental stresses, such as high radiation and temperature and low water availability, typical in areas with Mediterranean-type climate, cause an imbalance between the generation of electrons and their utilization. Reaction centers are unable to accept another electron, and the excitation energy or electrons lead to increased generation of reactive oxygen species [43], which cause considerable changes in the quantum yield of PSII. Whereas sclerophyllous species have adaptations to cope with these abiotic stresses, the responses of newly established seedlings also depend on the availability of water during the summer months. In our study, PI_ABS_, one of the most important parameters of chlorophyll a fluorescence related to plant vitality [44], together with F_V_/F_M_, F_V_/F_0_, and ψE_0_, were only affected by irrigation in *L*. *caustica* and *P*. *boldus*. *P*. *boldus* decreased PI_ABS_ and ψE_0_ in W−, but interestingly, *L*. *caustica* exhibited the opposite behavior in all four parameters. This implies that this species was not severely affected in the W− treatment as a consequence of its high ability to tolerate water deficit. This was demonstrated by Peña-Rojas et al. [45], who reported similar F_V_/F_M_ at predawn and photochemical efficiency of PSII (Φ_PSII_) in seedlings cultivated with 19% relative water content in comparison to those cultivated with 85% relative water content. This might be a consequence of the leaves’ angle and folding, which improves light capture [46]. It has been reported that electron transport is severely inhibited at a very low relative water content (e.g., 36% of water holding capacity) and high plant water potentials [23,47]. Di Marco et al. [48] observed a lack of damage to PSII in moderately stressed leaves of hard wheat grown in the field (water potentials from −1.0 to −4.0 MPa). Similarly, Fini et al. [49] and Swoczyna et al. [50] observed that F_V_/F_M_ did not respond to drought in tolerant tree species, and Peguero-Pina et al. [25] found that F_V_/F_M_ was reduced to ca. 0.3–0.4 when *Quercus* species experienced predawn water potential close to −7 MPa. The midday water potentials of −2.8 and −1.3 MPa in the seedlings of *P*. *boldus* and *L*. *caustica* suggest that the PSII of *P*. *boldus* was affected by the irrigation treatments imposed, which did not cause severe damage in *L*. *caustica*. This suggests that the magnitude of changes in quantum yield of PSII in our study was mainly species-dependent [51] and not exclusively dependent on the irrigation treatments. This may be explained by the fact that water renewal events occurred every 7 days during the period of greatest drought stress, when water was too low (2 L^−1^) for plant use. In the W+ treatment, the soil water content fell from 0.26 cm^3^ cm^−3^ the day of watering to 0.18 cm^3^ cm^−3^ two days after irrigation, whereas this value averaged 0.15 cm^3^ cm^−3^ in the W− treatment. As mentioned above, the study site was completely burned in 2017, and this may have caused a decrease in soil water retention [52]. Moreover, high temperatures during summer (average 30 °C) caused water to evaporate in a couple of days, which probably confounded the effects of irrigation and needs to be further investigated in terms of doses and frequency of irrigation.

Our results confirm that the technique of chlorophyll a fluorescence is a useful tool to assess plant responses to planting stress, providing information to support restoration projects in environments with a Mediterranean-type climate characterized by restricted water availability and high temperatures during the summer months. In our study, newly established seedlings of all the species, but particularly *P*. *boldus*, could benefit from providing additional protection from sunlight. It has been observed that the use of nurse plants (i.e., an adult plant whose close spatial association with the seedlings of young plants of another species has a positive effect on them) facilitates the establishment of Mediterranean species in dry sites, as they reduce the negative effects of excessive irradiation and water scarcity [53,54]. In a study on Mediterranean species, Ceacero et al. [55] found that protected plants showed higher photochemical efficiency than unprotected plants. Thus, excess light might also be an important factor affecting the initial establishment of the species under study, and the success of their restoration will depend on managing light intensity at the early stages of establishment.

## 4. Materials and Methods

### 4.1. Sites

The trial was established at an experimental station property of the Universidad de Chile located in central Chile (35°34′ S, 72°06′ W, 254 m a.s.l). In 2017, the total area of the experimental station was consumed by fire. Prior to the fire, the vegetation at the study site was compounded by (1) scattered sclerophyllous trees and shrubs, (2) young *Pinus radiata* D. Don plantations, and (3) pasture grasses. The climate at the study site is Mediterranean-type. The mean monthly precipitation and maximum temperature for the study period are shown in Figure 5, which were obtained from a weather station located 15 km south of the planting site and property of the Instituto de Investigaciones Agropecuarias (“https://agrometeorologia.cl (accessed on 20 December 2024)”). The vapor pressure deficit (VPD) in the summer months reached maximum daily values close to 3.0 kPa. The soil at the study site is neutral (pH 6.1), sandy clay (47% sand, 17% lime, 36% clay) with low electrical conductivity (0.03 dS m^−1^) and 1.5% organic matter content.

### 4.2. Species

The main characteristics and visual representations of *Q*. *saponaria*, *V*. *caven*, *L*. *caustica*, *E*. *pulverulenta*, *P*. *boldus,* and *C*. *odorifera* are provided in Table 3 and Appendix A. Seeds of the species were grown in a nursery property of the Forestal Arauco Company (35° 18′ S, 72° 23′ W, 10 m a.s.l), located in the city of Constitución, central Chile. The seeds were collected from December 2018 to May 2019, 20 km from the planting site, and a mix of seeds from different mother trees represented each species. In the case of *P*. *boldus*, the species has germination difficulties due to dormancy imposed by the essential oils of the pericarp [54]; thus, the seeds of this species were collected and immediately sown in January 2019 after excluding those that floated and manually removing the pulp. The seeds of the rest of the species were kept at 4° C until September 2019. After that, they were soaked in distilled water for 24 h and, after excluding those that floated, were set to germinate. Seedlings were grown in 140 mL pots filled with composted bark of *P*. *radiata* combined with the slow-release fertilizer Basacote 9M^®^ (COMPO GmbH & Co KG, Münster, Germany) at a dose of 3 g L^−1^ and cultured under ambient conditions of temperature and light until June 2020. At this stage, the seedling height was 20, 32, 25, 20, 25, and 35 cm for *P*. *boldus*, *C*. *odorifera*, *E*. *pulverulenta*, *V*. *caven*, *L*. *caustica*, and *Q*. *saponaria*, respectively.

### 4.3. Trial Establishment and Watering Treatments

The trial was established in July 2020, and the layout was a split-plot design with repeated measurements and five blocks. Two levels of watering (2 L plant^−1^ week^−1^ from November 2020 to March 2021 and from November 2021 to March 2022 (W+) (see Figure 1) and no watering (W−)) were the whole plot treatment, and the six species (*P*. *boldus*, *E*. *pulverulenta*, *Q*. *saponaria*, *L*. *caustica*, *C*. *odorifera*, and *V*. *caven*) were the split-plot treatment. Soil water content was measured using a soil moisture sensor (GS3 dielectric probe, Decagon Devices Inc., Pullman, WA, USA), which averaged 0.26 and 0.18 cm^3^ cm^−3^, respectively, on the same day and two days after irrigation of the W+ treatment, whereas this value averaged 0.15 cm^3^ cm^−3^ in the W− treatment. Average midday water potential, as measured with a pressure pump in a subsample of three seedlings per species per treatment, was −1.9 and −2.7 MPa in W+ and W−, respectively. The experimental unit was a rectangular plot of 10 seedlings in a 2 × 5 m seedling arrangement. The total number of seedlings was 600 (i.e., 2 watering treatments × 6 species × 5 blocks × 10 seedlings per plot), and there were no buffer rows between plots with different species. The whole plots were buffered by a row of *L*. *caustica* seedlings planted at a spacing of 1 × 1 m and were separated 20 m from each other to avoid cross-over of the watering effect. The seedlings were hand-planted in 40 × 40 × 30 cm planting holes at a spacing of 1 × 1 m (i.e., tree stocking of 10,000 stems ha^−1^). No fertilizer was added to the planting holes. The planting site was fenced, and the seedlings were protected from animal damage with tree shelters (Bioland S.A., Santiago, Chile), which were triangular, green, polyethylene tubes (45 cm tall).

### 4.4. Data Collection and OJIP Tests

Two years after establishment, Chl a fluorescence transient of intact and healthy leaves of similar size in the six species under study was measured at 10.00 h local time with a modulated fluorimeter (OSp30+, Optisciences, Hudson, NH, USA) set for the OJIP-test protocol. The measurements were performed in a sample of three seedlings (1 leaf per seedling) in 3 blocks in both the W+ and W− treatments (i.e., a total of 108 seedlings) on 4 and 18 December 2021, 8 January 2022, 7 February 2022, 8 March 2022, and 11 April 2022. The same leaves and plants were measured on each date. The leaves were dark-adapted in clips for 30 min prior to measurements, and later, Chl a fluorescence transients of the dark-adapted leaves were measured. The transients were induced by 1 s illumination, providing a maximum light intensity of 3500 μmol (photon) m^–2^ s^–1^. This light intensity was safe and allowed us to reach F_M_, and the OJIP seps were clearly revealed. The OSp30+ fluorimeter was at O (20 µs), J (2 ms), and I (30 ms) as the intermediate stage, and P (300 ms) as the peak. The data obtained were used in the OJIP-test [16] to calculate the parameters of photosystem II (PSII) photochemistry. With respect to the whole list of the OJIP-test parameters [12,16], we focused on the key parameters that are the major plant stress indicators [5,11], i.e., the maximum quantum yield of primary PSII photochemistry (F_V_/F_M_), the normalized area below the OJIP curve (S_m_), the apparent antenna size of an active PSII (ABS/RC), the ratio between variable and minimal fluorescence (F_V_/F_0_), the probability that a photon trapped by the PSII reaction center enters the electron transport chain (ψE_0_), the efficiency with which a PSII trapped electron is transferred to final PSI acceptors (ΔV_IP_), and the potential for energy conservation from photons absorbed by PSII to the reduction of intersystem electron acceptors (PI_ABS_). The rest of the variables related to phenomenological fluxes (per cross-section, CS) and specific fluxes (per reaction center, RC) were excluded from the analyses.

### 4.5. Data Analysis

Data were evaluated through variance analysis comparing treatments (W+ and W− treatments), species (*P*. *boldus*, *E*. *pulverulenta*, *Q*. *saponaria*, *L*. *caustica*, *C*. *odorifera*, and *V*. *caven*), dates (December 2021 to April 2022), and the interaction among factors. All factors were considered fixed, and significant values means were separated by Tukey’s test for *p* < 0.05. Mauchly’s test of sphericity was used to test equality of variances. The model terms were as follows:Y = µ + B + W + e_1_ +S + W × S + e_2_ + D + W × D + S × D + W × S × D + e_3_(1)
where Y is the observed phenotypic value, µ is the overall mean, B is the random effect of block, W is the fixed effect of watering regime, S is the fixed effect of species, D is the fixed effect of date, e_1_ is the error associated with the whole plot, e_2_ is the error associated with the split-plot, and e_3_ is the error associated with the measurement of the same individuals across dates. All the statistical analyses were performed with SPSS version 22.0 software (SPSS Inc., Chicago, IL, USA).

## 5. Conclusions

The comparative observations of our study demonstrate that differences in the photochemical efficiency of PSII vary markedly among the species under study and seem to be associated with foliar adaptations of the species more than the irrigation treatment. The evergreens *Q*. *saponaria*, *L*. *caustica*, and *E*. *pulverulenta* seem to have minor variations in PSII photochemical efficiency, whereas the deciduous and semi-deciduous *V*. *caven* and *C*. *odorifera* take advantage of their foliar habit and exhibit the highest photosynthetic performance. *P*. *boldus*, despite being an evergreen tree with leaf folding, seems to be more prone to photoinhibition by high summer temperatures. These findings can guide the restoration of Mediterranean-type climate areas by choosing species with adaptations to cope with high radiation loads experienced in these ecosystems. All the species studied, except *P*. *boldus*, have the capacity to survive and successfully establish after dry summers in Mediterranean-type climates. The successful restoration of *P*. *boldus* requires more attention to factors such as quality of the planting stock, irrigation dose, and additional protection to summer irradiance (e.g., nurse plants).

## Figures and Tables

**Figure 1 plants-14-02682-f001:**
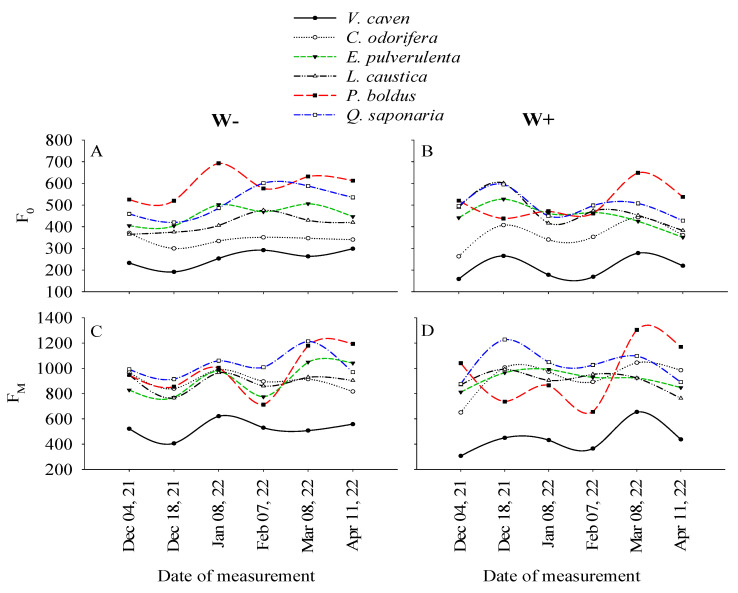
Values of F_0_ (minimal fluorescence from a dark-adapted leaf; **A**,**B**) and F_M_ (maximal fluorescence from a dark-adapted leaf; **C**,**D**) of the six sclerophyllous species grown under the well-watered (W+) and water-restricted (W−) treatments during the study period. Each symbol represents the average values for nine seedlings on each measurement date and watering treatment.

**Figure 2 plants-14-02682-f002:**
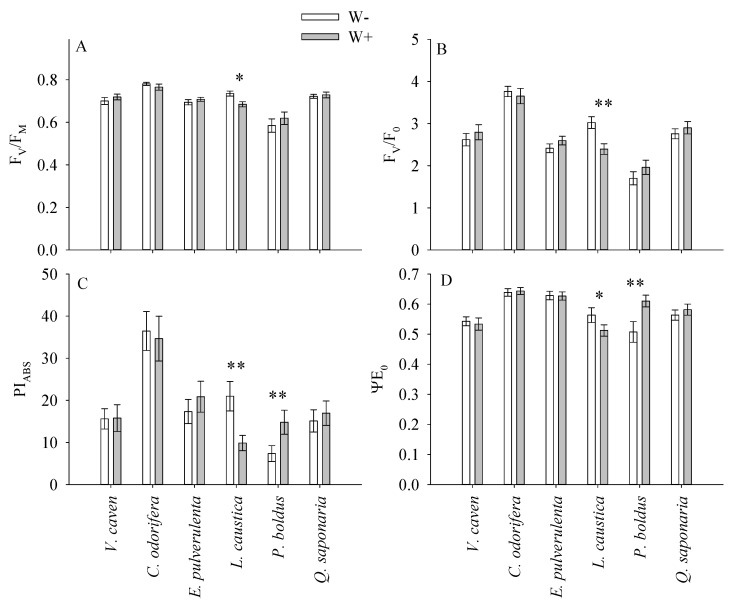
Responses of the chlorophyll fluorescence parameters of the different species according to the watering treatment. F_V_/F_M_ = maximum quantum yield of primary PSII photochemistry (**A**), F_V_/F_0_ = ratio between variable and minimal fluorescence (**B**), PI_ABS_ = potential for energy conservation from photons absorbed by PSII to the reduction of intersystem electron acceptors (**C**), ψE_0_ = probability that a photon trapped by the PSII reaction center enters the electron transport chain (**D**). W− and W+ represent the water-stressed and well-watered treatments, respectively. Asterisks indicate significant differences for a species in both the W+ (well-watered) and W− (water-restricted) treatments (0.01 and 0.05 for ** and *). Each bar represents average values for nine seedlings across dates of measurement.

**Figure 3 plants-14-02682-f003:**
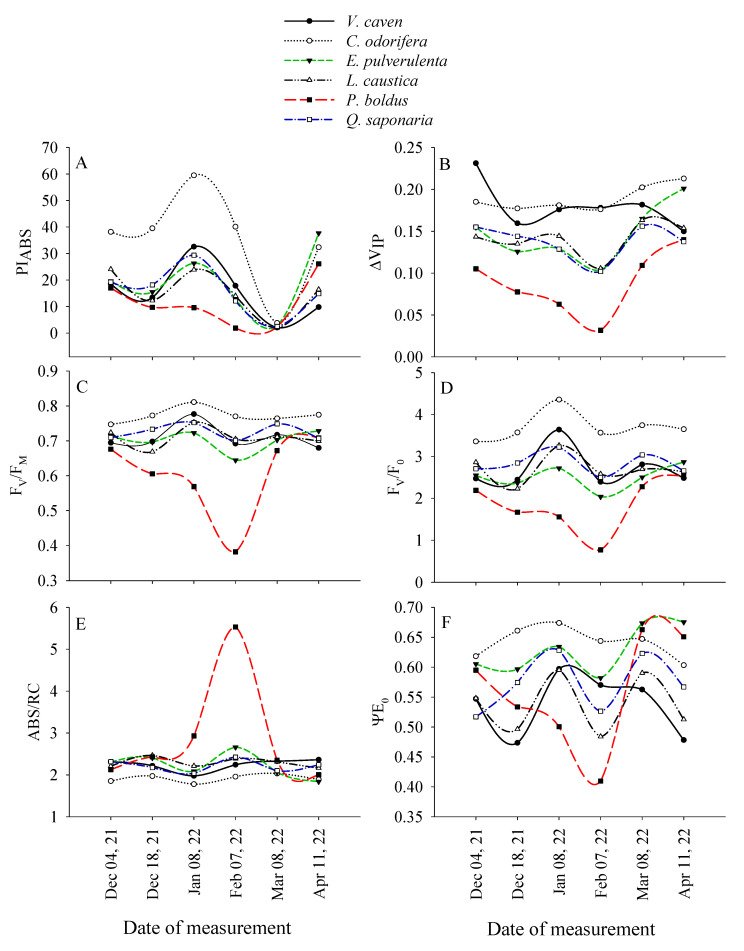
Temporal variations in chlorophyll fluorescence parameters among species. PI_ABS_ = potential for energy conservation from photons absorbed by PSII to the reduction of intersystem electron acceptors (**A**), ΔV_IP_ = efficiency with which a PSII trapped electron is transferred to final PSI acceptors (**B**), F_V_/F_M_ = maximum quantum yield of primary PSII photochemistry (**C**), F_V_/F_0_ = ratio between variable and minimal fluorescence (**D**), ABS/RC = apparent antenna size of an active PSII (**E**), and ψE_0_ = probability that a photon trapped by the PSII reaction center enters the electron transport chain (**F**). Each symbol represents average values for nine seedlings on each date of measurement and watering treatment.

**Figure 4 plants-14-02682-f004:**
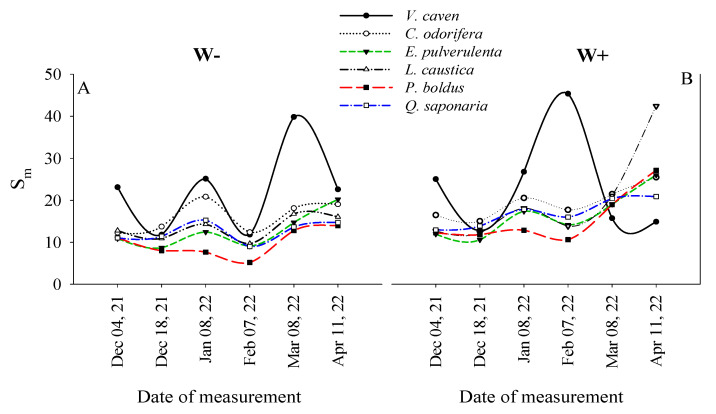
Interactive effects of species, watering, and date on the normalized area above the OJIP transient (S_m_). W− (**A**) and W+ (**B**) represent water-stressed and well-watered treatments, respectively. Each symbol represents average values for nine seedlings on each measurement date and watering treatment.

**Figure 5 plants-14-02682-f005:**
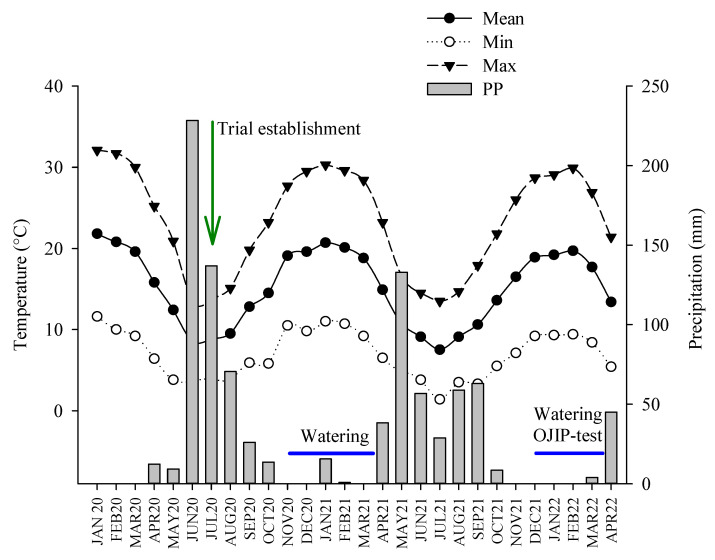
Monthly evolution of meteorological variables from planting to two years after outplanting. The inverted green arrow indicates the date at which seedlings were established in the field, and the blue lines indicate the implementation of watering in two seasons and OJIP measurements.

**Table 1 plants-14-02682-t001:** F-value and significance of F_0_ and F_M_ for the species and irrigations under study.

Source of Variation	F_0_	F_M_
Watering (W)	0.76 ns	0.49 ns
Species (S)	138.7 **	92.8 **
Date (D)	5.91 **	11.1 **
W × S	6.3 **	0.8 ns
W × D	10.0 **	4.7 **
S × D	1.4 ns	3.3 **
W × S × D	1.5 *	1.7 *

F_0_ = Minimal fluorescence from a dark-adapted leaf. F_M_ = Maximal fluorescence from a dark-adapted leaf. * and ** represent 0.05 and 0.01 significance levels, respectively, and ns is non-significant.

**Table 2 plants-14-02682-t002:** F-value and significance of the parameters of the JIP-test for the six species under study.

Source of Variation	F_V_/F_M_	S_m_	ABS/RC	F_V_/F_0_	PI_ABS_	ψE_0_	ΔV_IP_
Watering (W)	0.0 ns	23.7 **	3.5 ns	0.0 ns	0.0 ns	1.5 ns	52.4 **
Species (S)	44.9 **	11.5 **	11.7 **	56.6 **	33.8 **	18.1 **	57.8 **
Date (D)	12.0 **	12.2 **	9.1 **	12.3 **	39.3 **	9.6 **	17.5 **
W × S	3.2 **	0.8 ns	2.1 ns	4.4 **	4.5 **	5.8 **	1.3 ns
W × D	4.4 **	3.6 **	2.3 *	5.3 **	4.9 **	6.0 **	2.9 *
S × D	6.2 **	2.0 **	5.2 **	2.8 **	4.4 **	3.8 **	3.1 **
W × S × D	0.6 ns	2.8 **	1.0 ns	0.6 ns	1.1 ns	1.0 ns	1.0 ns

F_V_/F_M_ = maximum quantum yield of primary PSII photochemistry, S_m_ = normalized area above the OJIP transient, ABS/RC = apparent antenna size of an active PSII, F_V_/F_0_ = ratio between variable and minimal fluorescence, PI_ABS_ = potential for energy conservation from photons absorbed by PSII to the reduction of intersystem electron acceptors, ψE_0_ = probability that a photon trapped by the PSII reaction center enters the electron transport chain, ΔV_IP_ = efficiency with which a PSII trapped electron is transferred to final PSI acceptors. * and ** represent 0.05 and 0.01 significance levels, respectively, and ns is non-significant.

**Table 3 plants-14-02682-t003:** Main characteristics of the species under study.

Species	Life Form and Leaf Habit	Leaf Inclination Angle ^1^ and Characteristic of the Lamina	Leaf Absorptance (%) ^2^	References
*Q*. *saponaria*	tree, broad-leaved evergreen	45°, flat lamina	78.3	[20,21,27,28,29,30,31,32,33]
*C*. *odorifera*	shrub, semi-deciduous	65°, flat lamina	81.7	
*E*. *pulverulenta*	shrub, broad-leaved evergreen	45°, slightly rolled towards the abaxial surface	nd	
*L*. *caustica*	tree, broad-leaved evergreen	45°, lamina folded upwards (V-shaped)	87.2	
*V*. *caven*	shrub, deciduous	15°, small compound leaves	nd	
*P*. *boldus*	tree, broad-leaved evergreen	45°, rolled towards the abaxial surface	nd	

^1^: Leaf angle was measured with an inclinometer as the angle from the horizontal (0° ∼ flat; 90° ∼ vertical/steep) in two-year-old seedlings of the experiment. ^2^: Fraction of the incident photon or energy flux that is absorbed by the leaf. nd = no data.

## Data Availability

The data that support the findings of this study are available from the corresponding author upon reasonable request.

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
