# Peer review of "Foliar Traits Drive Chlorophyll Fluorescence Variability in Chilean Sclerophyllous Species Under Early Outplanting Stress"

_plants, 2025, doi:10.3390/plants14172682_

Round 1
Reviewer 1 Report
Comments and Suggestions for Authors
This study on Chilean sclerophyllous species provides valuable insights into the role of chlorophyll fluorescence parameters in understanding light protection mechanisms during early outplanting in a Mediterranean-type climate. The focus on how foliar adaptations and irrigation treatments influence these mechanisms is a promising avenue, particularly given the novelty of the approach. However, the limited evidence regarding the specific influence of foliar adaptations highlights a critical area for improvement.
Main comments:
- Please provide photos of different plant leaf shapes, which would more intuitively illustrate the adaptability of leaves to water scarcity.
- To strengthen the findings, a more detailed micro-observation of interspecific differences, such as the number and size of stomata; leaf tissue sections, etc.
- line 144-145, Asterisks indicate significant differences for a species in both W+ and W- treatments (0.01 and 0.05 for * and **). Please check the errors. * and ** represents 0.05 and 0.01 significance levels respectively.
- Why the seedlings were used in this study? Moreover, the manuscript lacks clarity on the specific age or developmental stage of the seedlings used, which is a critical detail for interpreting the results and ensuring reproducibility.
- P. boldus despite being an evergreen tree with leaf folding, appears more susceptible to photoinhibition due to high summer temperatures. Do its leaves fold during the seedling stage? This highlights the need to showcase the morphological traits of different tree species in this study, including both seedling and mature stages, to enhance reader understanding.
- Some journal abbreviations in the reference list are incorrect. e.g. line 406
Author Response
Please provide photos of different plant leaf shapes, which would more intuitively illustrate the adaptability of leaves to water scarcity.
Answer: Please see below a set of photos that illustrate the type of leaves of the different species in the trial. You can see the tiny compound leaves of V. caven (A), leaves of E. pulverulenta (B), leaf rolling in P. boldus (C), V-shaped leaves in L. caustica (D), leaves of Q. saponaria (E), and leaves of C. odorifera taken in adult individuals (F) which are similar to those of seedlings in our trial. We must clarify that photos were taken for dissemination activities and do not have the resolution for scientific purposes.
To strengthen the findings, a more detailed micro-observation of interspecific differences, such as the number and size of stomata; leaf tissue sections, etc.
Answer: We appreciate this comment by the reviewer. It would have been interesting to have sections of leaf tissues and analyze leaf anatomical adaptations of the species under study. Unfortunately, we do not have these types of observations.
Asterisks indicate significant differences for a species in both W+ and W- treatments (0.01 and 0.05 for * and **). Please check the errors. * and ** represents 0.05 and 0.01 significance levels respectively.
Answer: Corrected in Line 149.
Why the seedlings were used in this study? Moreover, the manuscript lacks clarity on the specific age or developmental stage of the seedlings used, which is a critical detail for interpreting the results and ensuring reproducibility.
Answer: As previously mentioned, we chose the species for their different adaptations in leaves (Line 89). We have now added a short paragraph indicating that these are the most relevant species in the ecosystem under study (Lines 88-89). We thank the comment and agree with the reviewer in the fact that the ontogenic stage of species is relevant to understanding the results. With respect to the developmental stage, it must be clarified that we worked with nursery seedlings cultured under homogenous conditions of light and irrigation and then established in the field under natural and stressful conditions. Please see this information in Lines 330-342. Seedling height at the moment of establishment was added in Lines 343-344 to illustrate the stage of development.
P. boldus despite being an evergreen tree with leaf folding, appears more susceptible to photoinhibition due to high summer temperatures. Do its leaves fold during the seedling stage? This highlights the need to showcase the morphological traits of different tree species in this study, including both seedling and mature stages, to enhance reader understanding.
Answer: Thank you for the comment. The answer is yes, the leaves of P. boldus experience rolling in the seedling stage (our study) and adult stage. Please see the reference by Doll et al. 2005 (in the number 29 of the reference list) showing leaf rolling in 6 year-old individuals and the following photograph of an adult individual.
Some journal abbreviations in the reference list are incorrect.
Answer: Corrected in Line 468. We also double-checked all references and corrected abbreviations in Lines 448 and 503

Reviewer 2 Report
Comments and Suggestions for Authors
- The current title is descriptive but could be more specific. Consider rephrasing to reflect the key finding, such as: “Foliar traits drive chlorophyll fluorescence variability in Chilean sclerophyllous species under early outplanting stress.”
- Table 3 lacks leaf absorptance values for P. boldus, V. caven, and E. pulverulenta. If unavailable, state so explicitly; if available from other literature, consider including.
-
More details on the frequency and timing of irrigation events (e.g., precise days or weeks) would help readers assess the experimental rigor. Also, specify whether irrigation mimicked natural rainfall patterns or was applied uniformly.
-
It is unclear whether the same leaves or plants were used repeatedly across measurement dates. Clarifying this is essential for understanding temporal trends and variability.
-
The use of repeated measures is appropriate, but the manuscript should clarify how assumptions (e.g., sphericity) were tested and whether corrections (e.g., Greenhouse-Geisser) were applied when necessary.
-
In Figures 1-4, the legends are minimal. Improve clarity by clearly labeling all axes and explicitly stating what each color or marker represents for each species and treatment.
-
The conclusions are somewhat descriptive. Please revise to include specific recommendations for restoration planning e.g., which species are best suited for which environments or conditions.
- Line 28–29: The statement “no clear effects of the irrigation treatments were observed among species and dates of measurement” contradicts later discussions (e.g., line 256–257). Please clarify or reconcile these discrepancies.
-
Line 36–60: The introduction is comprehensive but quite dense. Consider breaking long paragraphs to improve readability and better guide readers through the rationale and objectives.
-
Line 64–65: The sentence "i.e., sun at higher angle in the sky" is vague. Specify the time of day or season this refers to and its physiological implications more clearly.
-
Line 94–95: In the results, you mention that variations in F0 and FM were "mostly attributed to the effect of the species and date of measurement". Please quantify or illustrate this with exact F-values or effect sizes from Table 1 to support the claim.
-
Line 121–125: The interpretation of L. caustica's higher PIABS in W− is interesting but unexplained. Please expand on the possible physiological mechanisms that could explain improved performance under water restriction.
-
Line 149–151: The drastic drop in photosynthetic parameters for P. boldus during February is critical. Consider discussing how extreme temperature or vapor pressure deficits (Fig. 5) might have contributed.
-
Line 199–201: The sentence on leaf absorptance and angle effects could benefit from supporting citations or brief explanations for how these traits mechanistically influence fluorescence parameters.
-
Line 272–276: You mention rapid soil moisture loss due to high temperature, possibly confounding irrigation effects. Consider quantifying evaporation rates if data are available, or acknowledge this more clearly in the limitations.
Comments on the Quality of English Language
The manuscript communicates the scientific content effectively, there are several grammatical inconsistencies and awkward phrasings throughout the text which could be revised for smoother flow.
Author Response
The current title is descriptive but could be more specific. Consider rephrasing to reflect the key finding, such as: “Foliar traits drive chlorophyll fluorescence variability in Chilean sclerophyllous species under early outplanting stress.”
Answer: We appreciate the excellent reviewer’s suggestion. We have modified the title as suggested in Lines 2-3.
Table 3 lacks leaf absorptance values for P. boldus, V. caven, and E. pulverulenta. If unavailable, state so explicitly; if available from other literature, consider including.
Answer: Unfortunately, there is no information on this topic for the three species. For clarity, we replaced ‘-‘ by ‘nd’ in the body of table and provide the meaning of ‘nd’ in Line 348.
More details on the frequency and timing of irrigation events (e.g., precise days or weeks) would help readers assess the experimental rigor. Also, specify whether irrigation mimicked natural rainfall patterns or was applied uniformly.
Answer: As can be seen in Lines 351-355 we applied 2 liters of water per plant every 7 days (i.e., in a weekly basis) from November 2020 to March 2021. The 2 liters were applied uniformly per plant.
It is unclear whether the same leaves or plants were used repeatedly across measurement dates. Clarifying this is essential for understanding temporal trends and variability.
Answer: We measured the same plant and leaves on each date (Line 377). We added ‘plants’ for clarity.
The use of repeated measures is appropriate, but the manuscript should clarify how assumptions (e.g., sphericity) were tested and whether corrections (e.g., Greenhouse-Geisser) were applied when necessary.
Answer: Thank you very much for your concern about our statistical analysis. In our case, the assumption about sphericity (equality of variances of the differences between conditions) was not violated (Mauchly’s test of sphericity), thus corrections were not necessary. Please see Line 400.
In Figures 1-4, the legends are minimal. Improve clarity by clearly labeling all axes and explicitly stating what each color or marker represents for each species and treatment.
Answer: Thanks for the suggestion. In Figure 1 we added ‘Date of measurement’ in X-axis. In this figure, the meaning of F0 and FM were added in the caption. The meaning of W+ and W- is already indicated in the caption. In Figure 2 we added the meaning of W+ and W- in the caption. In Figure 3, we added ‘Date of measurement’ in X-axis. In Figure 4 we added ‘Date of measurement’ in X-axis and the meaning of W+ and W- in the caption. In Figures 1, 3, and 4 the meaning of each color is stated in the upper legend, with different colors for each species. If we repeat this information in the caption, we believe it could be overladed and confusing. We respectfully believe that all this information makes all figures easily understandable. Please see comments by reviewer 4 in this topic.
The conclusions are somewhat descriptive. Please revise to include specific recommendations for restoration planning e.g., which species are best suited for which environments or conditions.
Answer: We added recommendations for restoration planning in Lines 418-422.
The statement “no clear effects of the irrigation treatments were observed among species and dates of measurement” contradicts later discussions (e.g., line 256–257). Please clarify or reconcile these discrepancies.
Answer: Thank you for this specific comment. We think there is no contradiction because the sentence in Line 269 refers to the interaction between irrigation × species (which had effects only in L. caustica and P. boldus), whereas the paragraph in Lines 28-29 extends the analysis to the triple interaction between irrigation × species × date.
The introduction is comprehensive but quite dense. Consider breaking long paragraphs to improve readability and better guide readers through the rationale and objectives.
Answer: We split several sentences and slightly changed redaction to improve readability (Lines 37-60).
The sentence "i.e., sun at higher angle in the sky" is vague. Specify the time of day or season this refers to and its physiological implications more clearly.
Answer: We added ‘midday’ that is when radiation is typically strongest and plants receive high heat-loads that can cause photoinhibition (Line 65).
In the results, you mention that variations in F0 and FM were "mostly attributed to the effect of the species and date of measurement". Please quantify or illustrate this with exact F-values or effect sizes from Table 1 to support the claim.
Answer: The F-values for F0 and FM in most of the sources of variation (not all of them) were the highest in species (S) and date (D). They were 138.7 and 92.8 for Species and 5.91 and 11.1 for D. Please see figures in Table 1.
Line 121–125: The interpretation of L. caustica's higher PIABS in W− is interesting but unexplained. Please expand on the possible physiological mechanisms that could explain improved performance under water restriction.
Answer: We discuss this topic in Lines 271-276.
The drastic drop in photosynthetic parameters for P. boldus during February is critical. Consider discussing how extreme temperature or vapor pressure deficits (Fig. 5) might have contributed.
Answer: We thank you for this comment. The effect of high temperatures on the photosynthetic apparatus of P. boldus is discussed in Lines 226-240. We slightly changed the Line 235 in this paragraph and added a call to Figure 5.
The sentence on leaf absorptance and angle effects could benefit from supporting citations or brief explanations for how these traits mechanistically influence fluorescence parameters.
Answer: We added a short paragraph for leaf absorptance in the Discussion (Lines 211-214). As we did not have values for leaf absorptance for all species, we did not deep the discussion of this topic.
Line 272–276: You mention rapid soil moisture loss due to high temperature, possibly confounding irrigation effects. Consider quantifying evaporation rates if data are available, or acknowledge this more clearly in the limitations.
Answer: We do not have precise data to assess evaporation rate. In Lines 288-296, we declared that irrigation treatment could have some limitations in our study due to the low amount of water applied per plant and the stressful environmental conditions of the study site during the summer months. We also declared the need for further research into these topics (Lines 295-296).
The manuscript communicates the scientific content effectively, there are several grammatical inconsistencies and awkward phrasings throughout the text which could be revised for smoother flow.
Answer: As the reviewer did not indicate in which lines there were those inconsistencies and awkward phrases, we double-checked English in the whole document. All minor changes were highlighted with tracked changes but no number of line.

Reviewer 3 Report
Comments and Suggestions for Authors
Literature context and novelty of the study (Introduction): It is worth highlighting more clearly the cognitive gap that this study seeks to fill. While the topic is important, it should be better exposed what exactly is novel about this approach and how it differs from previous work (e.g., are new physiological mechanisms being investigated? new stress conditions?).
No discussion of the potential role of UV-B: If light stress plays an important role in the study, it is also worth considering the potential impact of UV-B (especially in the context of climate change and ozone depletion), even if it was not directly measured.
No section on study limitations: The article does not include a section on methodological limitations. It would be worth adding a brief reflection on possible measurement errors, limited number of samples, field conditions or the influence of confounding factors.
Literature deficiencies or outdated citations: Some statements are not supported by current literature or are based on older sources. It is worthwhile to supplement them with more recent publications (e.g., from the last 5 years), especially where physiological mechanisms or species-specific responses are mentioned.
Author Response
Reviewer 3
Comments and Suggestions for Authors
Literature context and novelty of the study (Introduction): It is worth highlighting more clearly the cognitive gap that this study seeks to fill. While the topic is important, it should be better exposed what exactly is novel about this approach and how it differs from previous work (e.g., are new physiological mechanisms being investigated? new stress conditions?).
Answer: We appreciate this comment by the reviewer. We slightly changed the paragraph in Lines 66-69 to make highlight the novelty of our study.
No discussion of the potential role of UV-B: If light stress plays an important role in the study, it is also worth considering the potential impact of UV-B (especially in the context of climate change and ozone depletion), even if it was not directly measured.
Answer: We appreciate this comment by the reviewer. Respectfully, we believe this topic is out of the scope of our study.
No section on study limitations: The article does not include a section on methodological limitations. It would be worth adding a brief reflection on possible measurement errors, limited number of samples, field conditions or the influence of confounding factors.
Answer: Thanks for this comment. As mentioned in the answer for reviewer 2, the limitations of the irrigation treatment were recognized (Lines 288-296). We consider that there were no measuring errors because we always measured the same plant and the same leaf between 10.00 to 12.00 hrs during the study period.
Literature deficiencies or outdated citations: Some statements are not supported by current literature or are based on older sources. It is worthwhile to supplement them with more recent publications (e.g., from the last 5 years), especially where physiological mechanisms or species-specific responses are mentioned.
Answer: We recognize that some specific references for the species under study are outdated. The reason is because this topic was largely studied in the 80’s and then not studied for more than 40 years. We assume this study because the effects of climate change (heat waves, drought, high temperature) have stressed the necessity to provide new information that support future restoration projects of Mediterranean ecosystems that harbor the species we studied. This is why some specific references for the species under study are outdated and we believe that our results fill a gap largely understudied.

Reviewer 4 Report
Comments and Suggestions for Authors
This manuscript assessed PSII photochemical efficiency in six Chilean sclerophyllous species under contrasting irrigation regimes. The results indicate that species-specific foliar traits, rather than irrigation, predominantly influenced photosynthetic performance and photoprotection responses.
The manuscript is interesting and potentially suitable for publication after minor revisions. Figure captions should be expanded to include all relevant details, such as colour representations and sample sizes. It is recommended to discuss the potential limitations of using chlorophyll fluorescence curves compared to other physiological analysis methods.
The Discussion section would also benefit from the inclusion of future perspectives and comparisons with findings from other studies. Could the approach used be applied to other species?
In the Materials and Methods section, please indicate which software was used for data modelling and parameter calculation (including appropriate references). Additionally, provide details about the cultivation conditions or the origin of the plants analysed.
Author Response
The manuscript is interesting and potentially suitable for publication after minor revisions. Figure captions should be expanded to include all relevant details, such as colour representations and sample sizes. It is recommended to discuss the potential limitations of using chlorophyll fluorescence curves compared to other physiological analysis methods.
Answer: Thanks for these comments. By responding to reviewer 2 we improved figures 1, 3 and 4. Following suggestions of reviewer 4, we added a short explanation of sample size (Lines 117-118, 149, 170-171, 189-190). With respect to the limitations of using chlorophyll fluorescence measurements, we respectfully believe this is out of the scope of our objective. We aimed to analyze how foliar traits influence chlorophyll fluorescence variability and not to compare this information with measurements of gas exchange, carbon isotopic discrimination, or any other physiological method.
The Discussion section would also benefit from the inclusion of future perspectives and comparisons with findings from other studies. Could the approach used be applied to other species?
Answer: Thanks for this comment. At the end of the Discussion, we include a short paragraph addressing the suggestions (Lines 297-310). The approach of using chlorophyll a measurement used can be used to any species.
In the Materials and Methods section, please indicate which software was used for data modelling and parameter calculation (including appropriate references). Additionally, provide details about the cultivation conditions or the origin of the plants analysed.
Answer: As mentioned, we used SPSS version software. Please see Lines 406-407. With respect to the cultivation and origin of seeds, please refer to information in Lines 330-342.

Round 2
Reviewer 1 Report
Comments and Suggestions for Authors
The revised manuscript shows some improvement in the text. However, the author has not included visual representations of plant morphology, relying solely on textual descriptions, which makes it difficult for readers to intuitively perceive differences in the studied materials. Additionally, presenting seedling morphological characteristics through photographs, combined with the chlorophyll fluorescence parameters measured by the author, would be more convincing. While I acknowledge the significance of the author's research, the presentation of the results is insufficient.
Author Response
Reviewer: The revised manuscript shows some improvement in the text. However, the author has not included visual representations of plant morphology, relying solely on textual descriptions, which makes it difficult for readers to intuitively perceive differences in the studied materials. Additionally, presenting seedling morphological characteristics through photographs, combined with the chlorophyll fluorescence parameters measured by the author, would be more convincing. While I acknowledge the significance of the author's research, the presentation of the results is insufficient.
Answer: We acknowledge the second suggestion by the reviewer in this topic. To support the information provided in Table 3 and to make the differences in leaves among the species clearer, we have now included a supplementary file (Figure S1) with photographs of each species showing morphological adaptations of leaves. In the text this has been called in Lines 330-331 and Lines 425-426.
Round 3
Reviewer 1 Report
Comments and Suggestions for Authors
The revised manuscript has included images of different plant samples, which allows readers to intuitively observe the phenotypic differences among the materials. This addition also provides strong visual support for the authors’ conclusion that foliar traits drive chlorophyll fluorescence variability. I find the revision satisfactory and recommend that the manuscript be accepted for publication.